# A Descriptive Analysis of the Relationship between Social Media Use and Vaccine Hesitancy among a Sample of Unvaccinated Adults in Canada

**DOI:** 10.3390/vaccines10122096

**Published:** 2022-12-07

**Authors:** Thomas Yen-Ting Chen, Rachael Piltch-Loeb, Nigel Walsh Harriman, Marcia Testa, Elena Savoia

**Affiliations:** 1Emergency Preparedness Research, Evaluation & Practice Program, Harvard T.H. Chan School of Public Health, 677 Huntington Avenue, Boston, MA 02115, USA; 2Department of Medical Research & Education, Kaohsiung Veterans General Hospital, Kaohsiung City 813414, Taiwan; 3Department of Biostatistics, Harvard T.H. Chan School of Public Health, 677 Huntington Avenue, Boston, MA 02115, USA

**Keywords:** social media, COVID-19 vaccine, vaccine hesitancy, information channel, Canada

## Abstract

Both traditional and social media information sources have disseminated information on the COVID-19 pandemic. The content shared may influence public opinion on different mitigation strategies, including vaccination. Misinformation can alter risk perception and increase vaccine hesitancy. This study aimed to explore the impact of using social media as the primary information source about the COVID-19 vaccine on COVID-19 vaccine hesitancy among people living in Canada. Secondary objectives identified other predictors of vaccine hesitancy and distinguished the effects of using traditional and social media sources. We used quota sampling of adults in Canada [N = 985] to conduct an online survey on the Pollfish survey platform between 21st and 28th May 2021. We then used bivariate chi-squared tests and multivariable logistic regression modeling to explore the associations between using social media as one’s primary source of information about the COVID-19 vaccine and vaccine hesitancy. We further analyzed the association between specific types of channels of information and vaccine hesitancy. After controlling for covariates such as age, sex, race, and ethnicity, individuals reporting social media as their primary source of COVID-19 vaccine information versus those who had not used social media as their primary source of COVID-19 vaccine information had 50% higher odds of vaccine hesitancy. Among different channels of information, we found that information from television was associated with a 40% lower odds ratio for vaccine hesitancy. Since social media platforms play an essential role in influencing hesitancy in taking the COVID-19 vaccination, it is necessary to improve the quality of social media information sources and raise people’s trust in information. Meanwhile, traditional media channels, such as television, are still crucial for promoting vaccination programs.

## 1. Introduction

Despite the consensus among experts that the COVID-19 vaccination is a critical practice to limit the consequences of the COVID-19 pandemic, vaccine hesitancy continues [1]. In Canada, as of November 2022, COVID-19 vaccines are broadly available, yet 18% of the population (almost seven million) has not completed the initial protocol (two doses of an accepted COVID-19 vaccine, a mix of two accepted vaccines, or one dose of the Janssen/Johnson & Johnson vaccine) [2,3]. In addition to the primary vaccination series, a vaccination booster dose was incorporated in public health measures [4]. The National Advisory Committee on Immunization (NACI) now recommends offering a first booster dose of a COVID-19 vaccine ≥6 months after completion of a primary COVID-19 vaccine series to adults in Canada [5]. Meanwhile, vaccine hesitancy remains a difficult and complex issue. The prevalence and mortality rates of COVID-19 in Canada were generally lower when compared with the United States and European countries prior to November 2021 (cumulative cases per million people by 1 November 2021, US, 136,454; Europe, 87,865; and Canada, 44,811; cumulative deaths per million people by 1 November 2021, US, 2206; Europe, 1798; and Canada, 755; Figure 1) [6,7]. However, a substantial increase in cases after that time highlighted the need for continued public health responses and prevention efforts.

Factors influencing increased vaccine hesitancy constitute critical barriers to vaccine uptake, posing a severe threat to global health [8,9]. The World Health Organization (WHO) defined vaccine hesitancy as the “delay in acceptance or refusal of vaccines despite availability of vaccine services.” The construct of vaccine hesitancy is complex and context-specific, varying across time, place, and vaccines, and it is influenced by factors such as complacency, convenience, and confidence [10].

Understanding the determinants of COVID-19 vaccine hesitancy is critical for developing public health programs that improve vaccine acceptance and maximize immunization rates. In the early stages of the COVID-19 pandemic, researchers conducted surveys within priority populations to investigate potential determinants influencing intentions to receive the vaccine [11,12,13]. Additionally, some studies have investigated vaccination acceptance among the general population in Canada using surveys. Overall, 70% or more of respondents to these surveys have expressed some level of acceptance of the COVID-19 vaccine. Potential determinants of vaccine hesitancy that were identified included age, sex, race, education level, and working in a healthcare sector [14,15].

Public risk communication practices focusing on the vaccine’s effectiveness, safety, and availability have also been empirically investigated as an essential determinant of vaccine acceptance [16]. Information has been abundant since the start of the COVID-19 pandemic, with every media channel covering the latest pandemic developments [17]. Traditional and social media information channels distribute vaccination information and may be influential in affecting public opinion on whether people want to be vaccinated. TV, newspapers, and radio are examples of traditional media, while Facebook, Instagram, YouTube, Twitter, and Tik Tok are examples of social media. A study conducted by Charron et al. demonstrated that parents who received information from healthcare professionals were more likely to have their children receive the MMR and HBV vaccines than parents who received information from the Internet or relatives only [18]. Furthermore, de Vries et al. found that individuals with COVID-19 vaccine hesitancy used social media services such as WhatsApp more often, while non-hesitant respondents used traditional written media more often [19].

Moreover, during the COVID-19 epidemic, there has been information overload and information being exchanged extremely rapidly, which potentially raises anxiety and stress levels [20]. Mass media news coverage through the Internet often draws its content from a large number of scientific journal articles but often does not report or translate the findings correctly. Haneef et al. found at least one spin (misleading interpretation) in 114 (88%) of news items and 18 different types of spin in the news related to misleading reporting (59%), misleading interpretation (69%), or overgeneralization/misleading extrapolation (41%) of the results, such as extrapolating a beneficial effect from an animal study to humans (21%) [21,22]. In addition, easy access to sharing and downloading information has driven this information explosion, which has been facilitated by an ever-increasing number of smartphone/mobile phone users worldwide, which reached 6.65 billion in 2022 [23]. Therefore, the COVID-19 pandemic has spawned a parallel “info-demic,” in which numerous channels and digital media portals disseminate incorrect information and unsubstantiated health recommendations [24]. Misinformation about the pandemic as a whole has propagated online and increased after the vaccination was introduced [16]. Misinformation spread on the Internet can influence risk perception and vaccine apprehension, while correctly interpreted scientific evidence can convey the appropriate risk-to-benefit evidence [25].

Unlike traditional media, social media allows users to quickly generate and share material worldwide without needing editorial oversight, scientific review, and evidence-based evaluation. As a result, misinformation about vaccine efficacy and safety on such platforms can cause significant public health issues, including the potential for vaccine hesitancy or a loss of public trust in the development of the vaccine for the prevention of COVID-19 [16]. Researchers have demonstrated that exposure to vaccine-critical websites and blogs negatively impacts vaccination intentions [26]. Social media may spread misinformation effectively by using vivid storytelling and images [27]. However, social media users may represent a skewed demographic sample with pre-existing misconceptions about vaccination’s advantages and side effects and a lack of awareness about the repercussions of other vaccine-preventable diseases [28]. Compared with social media, there has been less discussion and research on the role of traditional media, such as television, newspapers, and radio, in spreading misinformation [29]. People receive information from a variety of sources during a pandemic [30,31]. Furthermore, media consumption habits are frequently established before a specific event or topic of interest [32,33]. According to a previous study examining COVID-19 vaccine hesitancy among the U.S. population, individuals who received information from traditional media rather than social media, or both traditional and social media were more likely to accept the vaccine [34]. However, there is limited data assessing the impact of traditional and social media information channels on the public’s perception of the COVID-19 vaccine in Canada.

Recognizing the critical role of social media in vaccine hesitancy allows healthcare leaders to develop effective public health interventions that are more responsive and attuned to misinformation spreading online, and it allows them to consider how social media platforms might structure content to support evidence-based communication about the risks and benefits of vaccination.

This study explored the impact of social media use and identified factors associated with COVID-19 vaccine hesitancy among people living in Canada. Our main objective was to describe the association between using social media as the primary information source about the COVID-19 vaccine, trust in information from the source, and vaccine hesitancy. The secondary objectives were to explore other predictors of vaccine hesitancy and elucidate the association between vaccine hesitance and the use of traditional and social media sources to guide future public health vaccination campaigns.

## 2. Materials and Methods

### 2.1. Study Design and Data Collection

We conducted a cross-sectional online survey using mobile devices on the Pollfish survey platform. Pollfish pays mobile application developers for displaying polls within their applications. To motivate participation, randomly selected users who complete the surveys are given non-monetary incentives such as an extra life in a game or access to premium content [35]. The Pollfish platform leverages random device engagement (RDE) to reach users who are solely recognized by a unique device ID while using a mobile application [36]. Pollfish has over 1 billion members globally, and the research team chose a random sample of Canadian users meeting the eligibility criteria described below. An initial survey draft underwent cognitive debriefing and usability testing by 20 individuals who spoke English and French, after which it was modified for length and clarity. The survey was designed to be completed on a mobile device within 10 min or less. The survey was conducted between 21st and 28th May 2021. After the first COVID-19 vaccine was authorized for use in Canada on 9 December 2020, healthcare workers and residents of long-term care institutions were given priority to be immunized [37]. Based on national guidance available at the time the survey was developed, populations with mortality risk factors, such as advanced age and comorbidities, were also included in the vaccine priority group [38]. Some provinces further gave priority to people who were at a higher risk of contracting the disease, such as essential frontline workers and residents of areas that were severely affected [39]. Individuals who resided in Canada were eligible to participate if they were over 18 years of age and had not yet been fully vaccinated (two doses of an accepted COVID-19 vaccine, a mix of two accepted vaccines, or one dose of the Janssen/Johnson & Johnson vaccine) at the time the survey was administered. The study protocol and survey instrument were approved by the Harvard T.H. Chan School of Public Health Institutional Review Board on 8 December 2020 (protocol #20-203). Participants reviewed the study’s information before agreeing to participate in the study by starting the survey and pressing “yes.” The English and French versions of the questionnaire are provided as Appendix A to this manuscript. To ensure its validity, the survey was translated into Canadian French and back-translated into English. To fully recognize the potential impact that the two different cultures may have on vaccine hesitancy, we took samples from two different groups in Canada: one from the English-speaking and one from the French-speaking. The sample was equally distributed by sex and age groups between French and English speakers. Respondents had the option of responding in English or French. The datasets derived from the two samples were analyzed independently. As three minutes was determined to be the minimal amount of time needed to thoughtfully complete the survey, we used this time threshold as a means for ensuring data quality and eliminated any questionnaires that were finished in under three minutes.

### 2.2. Dependent Variable (Vaccine Hesitancy)

Respondents were asked how likely they would be to receive a COVID-19 vaccine if offered to them at no cost within two months. The six response options were very likely, somewhat likely, somewhat unlikely, very unlikely, would consider it later on, and not sure. For the scope of this analysis, the six response options were dichotomized and coded as “vaccine-acceptant” if the answer “very likely” was chosen, and “vaccine hesitant” if any other category was chosen.

### 2.3. Independent Variables

The primary exposures of interest were whether a respondent relied on social media as one of the sources where they received the most information about the COVID-19 vaccine and their degree of trust in the source. Respondents were asked to choose up to three sources from which they had learned the most about the COVID-19 vaccine. Since different sources of information are not mutually exclusive, each one was regarded as an indicator variable that was dichotomized according to whether or not it had been chosen. We defined traditional media sources of information as TV, newspaper, or radio, and social media sources of information were defined as Facebook, Instagram, YouTube, Twitter, or Tik Tok. Trust in information was measured by analyzing responses to the question “How much do you trust the information you got so far about the COVID-19 vaccine?” Response options were categorized into three levels: (1) low trust—those that reported “not at all” or “very little”; (2) moderate trust—those that reported “somewhat”; and (3) high trust—those that reported “a lot.” Respondents were also asked how frequently they used social media. Response options were divided into four categories: (1) very low—those that reported “never”; (2) not often—those that reported “not often” or “alternate days”; (3) every day—those that reported “every day, for more than 1 h but less than 3 h”; and (4) high—those that reported “every day, for 3 h or more.”

In addition to the primary independent variables of interest, we also considered several covariates that have been linked to vaccine hesitancy and health behaviors including socio-demographic characteristics, experience with COVID-19, and perceived risk of contracting and spreading COVID-19. Socio-demographic variables included age, sex, provincial region, race, level of education, employment status, and type of job (working in the healthcare sector versus not working in this sector). A variable describing provincial regions of Canada was categorized into six groups: Western provinces (British Columbia, Alberta, Saskatchewan, and Manitoba), Ontario, Quebec, Atlantic provinces (Nova Scotia, New Brunswick, Prince Edward Island, and Newfoundland/Labrador), Northern provinces (Yukon, Northwest Territories, and Nunavut), and Other. COVID-19 experience included having been diagnosed with COVID-19 (yes/no) and knowing someone who had died of COVID-19 (yes/no). COVID-19 risk perception was measured by asking respondents to rate their level of concern of contracting COVID-19 at work or outside their work environment and their level of concern about infecting family members or friends. Responses were dichotomized into two risk-related categories (medium/low reported risk vs. high reported risk).

### 2.4. Statistical Analysis

In preparation for multivariable analysis and to answer our primary research question concerning COVID-19 vaccine hesitancy, we conducted bivariate analyses using chi-squared tests and multivariable logistic regression analysis to study the association between the independent variables and the odds of hesitancy in taking the vaccine. We also used descriptive statistics to analyze the proportion of respondents who used each information source, the level of trust in each channel, and the frequency with which they used social media.

We used chi-squared tests of independence to test for associations between the levels of each independent variable and the levels of the dependent variable indicative of vaccine acceptance or vaccine hesitancy. We used a *p*-value of <0.05 as the threshold for inclusion of the predictor variable in the multivariable logistic regression model. A Spearman’s rank correlation test was conducted to test for collinearity among predictors prior to the completion of the regression analysis, with the cutoff set at 0.7 as a measure of collinearity. For multivariable regression analysis, we conducted logistic regression to determine the odds ratio of being hesitant to receive the vaccine compared with those in the acceptance group. To evaluate if the association between information source and hesitancy varied by either trust in information or frequency of social media use, we added the interaction terms to the multivariable regression model. Significance was determined by a two-sided *p* < 0.05. A Hosmer–Lemeshow goodness-of-fit test for binary logistic regression models was conducted to assess the goodness of fit [40]. Data analysis was performed using Stata Statistical Software: Release 17 College Station, TX: StataCorp LLC.

## 3. Results

### 3.1. Sample Characteristics

There were 985 respondents in our analytic sample, and the detailed sample characteristics are described in Table 1. Among all respondents, 413 (41.9%) were hesitant to receive the vaccine in the next two months. The sample was predominantly White (69%), and 46% were from Quebec. The majority had a college/bachelor’s degree (58%), and 12% possessed a post-bachelor’s degree. For employment status, the majority of respondents (62%) reported they were currently working. In terms of COVID-19 experience, 12% of respondents reported having COVID-19, and 9% had a close family member or friend who had died of COVID-19 (Table 2). In terms of concern for COVID-19, 39% of respondents had a high level of perceived risk of contracting COVID-19 at work or outside of work, and 42% had a high level of perceived risk of infecting others with COVID-19.

### 3.2. Descriptive Statistics of COVID-19 Information from Different Channels

Table 3 demonstrates the proportion of users of each information channel stratified by the level of trust in information among channels as well as the frequency of using social media. The most frequently used source of COVID-19 vaccine information reported by respondents was TV (71.4%), followed by Facebook (34.2%), newspaper (26.5%), Twitter (20.0%), YouTube (17.9%), Instagram (13.6%), radio (12.1%), and Tik Tok (10.3%). Overall trust in vaccine information was “high” for 30.4% of respondents, “some” for 43.9%, and “low” for 25.8%. The level of trust in information was highest for radio (33.6% had high trust), newspapers (34.1%), and television (33.7%), while social media channels had lower frequencies of high trust (YouTube, 21.6%; Twitter, 25.4%; and Instagram, 26.1%). Overall, 35.2% of respondents reported high frequency of using social media (every day for 3 h or more) and 31.5% reported using social media every day. The percentage of users that reported “using social media every day, for 3 h or more” was highest for Tik Tok (58.4%), followed by Instagram (57.5%), YouTube (48.3%), Facebook (46.3%), and Twitter (45.7%). Those who received most of their information about the COVID-19 vaccine from social media reported a higher frequency of using social media than those who received most of their information from other sources (56.4% versus 29.2%). On the other hand, trust in information was higher among those who never used social media as the source to get the most information about the COVID-19 vaccine (high trust was 32.6% versus 22.5%).

### 3.3. Bivariate Analysis

Table 2 presents the bivariate analysis among variables and vaccine hesitancy. There were statistically significant differences (significance level of 0.05) in vaccine hesitancy by “having ever relied on social media as a primary source of information about the COVID-19 vaccine,” age, provincial region, race, education, employment, prior COVID-19 diagnosis, concern for infecting others with COVID-19, and level of trust in information. There was no significant difference in vaccination hesitancy based on sex, working in the healthcare sector, knowing someone who died of COVID-19, concern for contracting COVID-19, or frequency of using social media. Those who ever relied on social media as a primary source of COVID-19 vaccine information were more likely to be vaccine hesitant compared with those who had not (the percentage was 51.8 versus 39.1). In terms of different information channels, there was a statistically significant difference in vaccine hesitancy associated with getting information from TV. A higher proportion of individuals among those who had gotten information from TV (65.4%) were vaccine acceptant compared with those who had not.

### 3.4. Multivariable Regression Analysis

Table 4 displays the results of the multivariable logistic regression analyzing the effect of social media as a primary source of COVID-19 vaccine information on vaccine hesitancy. The Hosmer–Lemeshow goodness-of-fit test for binary logistic regression models confirmed the model was a good fit for the data (*p* = 0.998). We found that the odds of vaccine hesitancy were 50% higher among those who had used social media as a primary source of COVID-19 vaccine information (OR: 1.50, 95%CI: 1.03–2.19) compared with those who had not. Compared to those aged 18–24 years, respondents aged 45–54 years had 50% decreased odds of being vaccine hesitant (OR: 0.51, 95%CI: 0.31–0.85), respondents aged over 54 years had 60% decreased odds of being vaccine hesitant (OR: 0.40, 95%CI: 0.21–0.74), while other age subgroups did not present a significant difference. Compared with those in Western provinces (British Columbia, Alberta, Saskatchewan, and Manitoba), respondents in Quebec (OR: 1.69, 95%CI: 1.09–2.62) and the Northern provinces (OR: 10.59, 95%CI: 1.16–96.64) had 69% and 959% increased odds of being vaccine hesitant, respectively, while living in other provincial regions did not present a significant effect. We also observed a graded negative association between trust in information and vaccine hesitancy. Compared with those with low trust, respondents with some trust (odds ratio, OR: 0.25, 95% confidence interval, 95%CI: 0.17–0.36) and high trust in the information received (OR: 0.06, 95%CI: 0.04–0.10) had 75% and 94% decreased odds of being vaccine hesitant, respectively.

Finally, the analysis of trust in information and frequency of using social media as potential effect modifiers of the relationship between information source and vaccine hesitancy revealed no significant interaction effects between these variables. The association between social media use and vaccination hesitancy was consistent across individuals regardless of the level of trust in information and frequency of use of social media.

## 4. Discussion

Our study aimed to explore the impact of social media use and identify predictors of COVID-19 vaccine hesitancy among people living in Canada. Our results showed that 22.1% of respondents had relied on social media as a primary source of COVID-19 vaccine information and that this group had 1.5 times the odds of vaccine hesitancy as compared with those who had not relied on social media as a primary source of COVID-19 vaccine information.

We do not know if vaccine-hesitant individuals in our sample became hesitant due to exposure to information on social media or if they ended up using social media to get information about the COVID-19 vaccine because they were already vaccine-hesitant. A previous study conducted in Canada revealed that participants who would probably get vaccinated were more likely to report lower levels of mistrust toward authorities and higher perceived scientific consensus compared with those who were vaccine hesitant [41]. Interestingly, our results indicate that while trust in the source of information is associated with vaccine acceptance, those using social media as a primary source of information about the vaccine were hesitant regardless of their level of trust in the source. This means that even those who do not trust social media may still be influenced by the information they are exposed to in this environment. In addition, frequency of use of social media did not seem to make a difference, pointing to the possibility that anti-vax information may be effective even in small doses.

Regarding socio-demographic factors, we found respondents aged 45–54 years and over 54 years had significantly decreased odds of being vaccine hesitant when compared with those aged 18–24 years. Our results are consistent with previous studies in Canada and align with trends in the United States, which indicate that younger individuals (those under 65 years old) are less acceptant of the COVID-19 vaccine [42]. Dzieciolowska et al. found that being over 50 years old was a determinant of increased vaccine acceptance among healthcare workers in Canada [11]. Ogilvie et al. reported that individuals above the age of 60 were more likely to be vaccine acceptant [14]. Also, Tang et al. found that individuals under the age of 60 had lower vaccination intention than individuals above the age of 60 [15]. In general, younger populations are less willing to receive the COVID-19 vaccine, which may in part be explained by the way they obtain information and make health decisions. To promote vaccine acceptance in this population, it is imperative to have more targeted information which reassures them that the vaccine is both necessary and safe.

Regarding interprovincial comparisons, we found that those living in Quebec and the Northern provinces (Yukon, Northwest Territories, and Nunavut) had significantly higher odds of being vaccine-hesitant when compared with Western provinces (British Columbia, Alberta, Saskatchewan, and Manitoba). While inconsistent findings have been reported by previous investigations, and our results are not representative of all residents of these regions, what we found was comparable to those reported by Owen et al., where Western provinces were more likely to take the COVID-19 vaccine, followed by Ontario and Quebec [43]. However, Atlantic Canada led in willingness to vaccinate in that report, whereas the effect of living in Atlantic provinces and living in Ontario were not significantly different in our study. The findings were intriguing because Quebec was one of the provinces most afflicted by COVID-19 in terms of the number of cases and deaths, while it had higher odds of being vaccine hesitant [44]. According to a large population-based study in Quebec, speaking French at home was associated with vaccine hesitancy [45]. In our study population, out of the 449 participants living in Quebec, 409 (91.1%) were French speaking. It is conceivable that some cultural factors might have contributed to vaccine hesitancy.

On the contrary, another study found that rates of COVID-19 vaccine hesitancy were higher among those living in Western provinces and Ontario compared with Quebec and the Atlantic provinces [46]. Although living in the Northern provinces had an odds ratio point estimate of 10.59, this finding was based on only one vaccine-acceptant respondent and had wide confidence intervals. While more large-scale, and representative studies are needed to determine interprovincial differences, the fact that vaccine hesitancy differs between settings necessitates local assessments to develop tailored vaccine campaigns [47].

Traditional media channels are important for promoting vaccination programs since they are likely to use high-quality sources and present fact-based vaccine information supported by governmental, healthcare, or academic data and studies [48]. Furthermore, due to their widespread use, social media platforms can play a pivotal role in educating those who are most hesitant because these social media platforms might attract vaccine-hesitant individuals. By evaluating survey respondents’ self-reported social media use and their level of acceptance of the COVID-19 vaccination, this study adds to the developing knowledge concerning vaccine hesitancy in relation to COVID-19 vaccines. A practical implication derived from our findings is that public health agencies need to work closely with social media companies to identify opportunities to reach vaccine-hesitant individuals using different types of platforms and algorithms. For example, they could work to create posts that will more effectively engage the public, since they are regarded as authoritative, reputable information sources and are likely to cause attitude and behavior changes regarding vaccine endorsement [49]. Public health agencies should launch focused campaigns on social media platforms in populations where hesitancy typically occurs [50]. Additionally, communication strategies should be tailored to each social media platform since these platforms vary in the degree of vaccine misinformation they disseminate to the public. On the other hand, social media platforms could help by extending vaccine-promoting organizations’ reach, boosting cooperation to promote credible content [51]. Rather than simply marking content as false or misinformation and removing or restricting the distribution of content, there is also an opportunity for these social networks to provide explanations and enlighten their users with reliable information [49,52].

Our study has several limitations. First, this is a cross-sectional study and is not able to determine the temporality of the observed relationships; as such, our reported associations should not be interpreted as causal. Second, because Pollfish was used, the sample was skewed toward people who most frequently use a mobile device. Third, traditional channels may be promoted or shared on social media channels, and the respondents were not asked to select a particular channel exclusively. The influence of this exchange cannot be separated in our analysis. This study also did not evaluate the cumulative effect of receiving information from multiple channels. Fourth, interprovincial comparisons are challenging because the number of participants was smaller in some provinces (e.g., Atlantic and Northern). In addition, the prevalence of COVID-19 in our sample was higher than the national average, indicating the sample’s expected greater exposure and infection rate. However, our study relied on self-reports of COVID-19 rather than laboratory-confirmed cases. Furthermore, we were unable to validate our findings in terms of the impact of the variables we discovered to be associated with intention to get vaccinated compared to actual behaviors because we are unsure of the degree to which expressed intent to take a vaccine is associated with actual behavior. Finally, this sample was not a representative sample of Canada, and the findings may not be generalized beyond this sample. Future research is still needed to determine the longitudinal impact of social media use.

## 5. Conclusions

In conclusion, our study demonstrated that the respondents in Canada who had ever relied on social media as a primary source of COVID-19 vaccine information had significantly higher odds of vaccine hesitancy. Since social media platforms play an important role in COVID-19 vaccine hesitancy, it is necessary to improve the quality of social media information sources and raise people’s trust in information.

## Figures and Tables

**Figure 1 vaccines-10-02096-f001:**
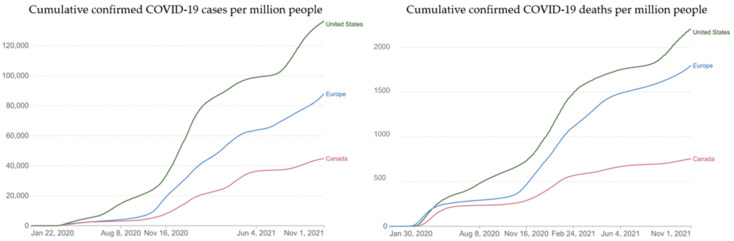
Cumulative confirmed COVID-19 cases and deaths in Canada, the United States, and Europe prior to November 2021. (Accessed through Our World in Data [OurWorldInData.org]. Available from: https://ourworldindata.org/covid-vaccinations. Source: Johns Hopkins University CSSE COVID-19 Data) (Accessed on 30 November 2022).

**Table 1 vaccines-10-02096-t001:** Levels of Vaccine Hesitancy and Baseline Demographic Characteristics of Respondents: Total and by Reporting Using Social Media as Their Primary Source of COVID-19 Vaccine Information.

Characteristics	Primary Source of COVID-19 Vaccination Information			*p*-Value
	Total n = 985	Not Social Media n = 767	Social Media n = 218	
Subtotals	Within Level of Characteristic (%)	Subtotal by Social Media Use (%)		
How likely to receive COVID-19 vaccine				
Very likely	572 (58%)	467 (61%)	105 (48%)	
Somewhat likely	140 (14%)	99 (13%)	41 (19%)	
Somewhat unlikely	31 (3%)	24 (3%)	7 (3%)	
Very unlikely	83 (8%)	54 (7%)	29 (13%)	
Would consider later on	46 (5%)	39 (5%)	7 (3%)	
Not sure	113 (11%)	84 (11%)	29 (13%)	
Age				<0.001
18–24	198 (20%)	129 (17%)	69 (32%)	
25–34	193 (20%)	134 (17%)	59 (27%)	
35–44	196 (20%)	152 (20%)	44 (20%)	
45–54	199 (20%)	171 (22%)	28 (13%)	
Over 54	199 (20%)	181 (24%)	18 (8%)	
Sex (female)	494 (50%)	357 (47%)	137 (63%)	<0.001
Healthcare sector	85 (9%)	64 (8%)	21 (10%)	0.55
Provincial region				0.092
Western †	199 (20%)	151 (20%)	48 (22%)	
Ontario	260 (26%)	192 (25%)	68 (31%)	
Quebec	449 (46%)	357 (47%)	92 (42%)	
Atlantic ‡	60 (6%)	50 (7%)	10 (5%)	
Northern ϕ	11 (2%)	11 (1%)	0 (0%)	
Other	6 (2%)	6 (1%)	0 (0%)	
Race				<0.001
White	677 (69%)	560 (73%)	117 (54%)	
Black	52 (5%)	37 (5%)	15 (7%)	
Asian	128 (13%)	78 (10%)	50 (23%)	
Hispanic	14 (1%)	12 (2%)	2 (1%)	
Other	114 (12%)	80 (10%)	34 (16%)	
Education				0.25
Less than high school	49 (5%)	38 (5%)	11 (5%)	
High school	238 (24%)	188 (25%)	50 (23%)	
College/Bachelor’ degree	572 (58%)	435 (57%)	137 (63%)	
Post-graduate	118 12%)	98 (13%)	20 (9%)	
Other	8 (1%)	8 (1%)	0 (0%)	
Employment				<0.001
Working	615 (62%)	476 (62%)	139 (64%)	
Not working	219 (22%)	159 (21%)	60 (28%)	
Retired	122 (12%)	112 (15%)	10 (5%)	
Other	29 (3%)	20 (3%)	9 (4%)	

† Western provinces: British Columbia, Alberta, Saskatchewan, and Manitoba. ‡ Atlantic provinces: Nova Scotia, New Brunswick, Prince Edward Island, and Newfoundland/Labrador. ϕ Northern provinces: Yukon, Northwest Territories, and Nunavut.

**Table 2 vaccines-10-02096-t002:** Univariate and Bivariate Analysis.

Covariate	Total n = 985 (100%)	Acceptant n = 572 (58.1%)	Hesitant n = 413 (41.9%)	*p*-Value
Subtotals	Within Level of Covariate (%)	Subtotal by Hesitancy (%)		
Most information channel				<0.001
Not social media	767 (77.9%)	467 (60.9%)	300 (39.1%)	
Social media	218 (22.1%)	105 (48.2%)	113 (51.8%)	
Age				<0.001
18–24	198 (20.1%)	93(47.0%)	105 (53.0%)	
25–34	193 (19.6%)	90 (46.6%)	103 (53.4%)	
35–44	196 (19.9%)	98 (50.0%)	98 (50.0%)	
45–54	199 (20.2%)	128 (64.3%)	71 (35.7%)	
Over 54	199 (20.2%)	163 (81.9%)	36 (18.1%)	
Sex (female)	494 (50.2%)	286 (57.9%)	208 (42.1%)	0.91
Healthcare sector	85 (8.6%)	45 (52.9%)	40 (47.1%)	0.32
Provincial region				0.026
Western †	199 (20%)	121 (21%)	78 (19%)	
Ontario	260 (26%)	159 (28%)	101 (25%)	
Quebec	449 (46%)	260 (46%)	189 (46%)	
Atlantic ‡	60 (6%)	27 (5%)	33 (8%)	
Northern ϕ	11 (1%)	1 (0%)	10 (2%)	
Other	6 (1%)	4 (1%)	2 (1%)	
Race				<0.001
White	677 (68.7%)	412 (60.9%)	265 (39.1%)	
Black	52 (5.3%)	24 (46.2%)	28 (53.9%)	
Asian	128 (13.0%)	86 (67.2%)	42 (32.8%)	
Hispanic	14 (1.4%)	5 (35.7%)	9 (64.3%)	
Other	114 (11.6%)	45 (39.5%)	69 (60.5%)	
Education				<0.001
Less than high school	49 (5.0%)	18 (36.7%)	31 (63.3%)	
High school	238 (24.2%)	129 (54.2%)	109 (45.8%)	
College/Bachelor’ degree	572 (58.1%)	353 (61.7%)	219 (38.3%)	
Post-graduate	118 (12.0%)	71 (60.2%)	47 (39.8%)	
Other	8 (0.8%)	1 (12.5%)	7 (87.5%)	
Employment				<0.001
Working	615 (62.4%)	367 (59.7%)	248 (40.3%)	
Not working	219 (22.2%)	87 (39.7%)	132 (60.3%)	
Retired	122 (12.4%)	103 (84.4%)	19 (15.6%)	
Other	29 (2.9%)	15 (51.7%)	14 (48.3%)	
COVID-19 experience				
Diagnosed with COVID-19	119 (12.1%)	42 (35.3%)	77 (64.7%)	<0.001
Knows someone who died	86 (8.7%)	53 (61.6%)	33 (38.4%)	0.48
Concern for COVID-19				
Contracting COVID-19	383 (38.9%)	235 (61.4%)	148 (38.7%)	0.095
Infecting others	414 (42.0%)	276 (66.7%)	138 (33.3%)	<0.001
Frequency of using social media				0.17
Very low	67 (6.8%)	37 (55.2%)	30 (44.8%)	
Not often	261 (26.5%)	151 (57.9%)	110 (42.2%)	
Every day	310 (31.5%)	195 (62.9%)	115 (37.1%)	
High	347 (35.2%)	189 (54.5%)	158 (45.5%)	
Trust in information				<0.001
Low	254 (25.8%)	59 (23.2%)	195 (76.8%)	
Some	432 (43.9%)	257 (59.5%)	175 (40.5%)	
High	299 (30.4%)	256 (85.6%)	43 (14.4%)	
Information from channel				
Facebook	337 (34.2%)	187 (55.5%)	150 (44.5%)	0.24
Twitter	197 (20.0%)	100 (50.8%)	97 (49.2%)	0.02
YouTube	176 (17.9%)	92 (52.3%)	84 (47.7%)	0.085
Instagram	134 (13.6%)	69 (51.5%)	65 (48.5%)	0.097
Tik Tok	101 (10.3%)	52 (51.5%)	49 (48.5%)	0.16
Television	703 (71.4%)	460 (65.4%)	243 (34.6%)	<0.001
Newspaper	261 (26.5%)	160 (61.3%)	101 (38.7%)	0.22
Radio	119 (12.1%)	70 (58.8%)	49 (41.2%)	0.86

† Western provinces: British Columbia, Alberta, Saskatchewan, and Manitoba. ‡ Atlantic provinces: Nova Scotia, New Brunswick, Prince Edward Island, and Newfoundland/Labrador. ϕ Northern provinces: Yukon, Northwest Territories, and Nunavut.

**Table 3 vaccines-10-02096-t003:** Description of COVID-19 Information from Different Channels.

	Users n (%)	Level of Trust in Vaccine Information from Channel n (%)			Frequency of Using Social Media n (%)			
		Low	Some	High	Very Low	Not Often	Every Day	High
Overall	985 (100%)	254 (25.8%)	432 (43.9%)	299 (30.4%)	67 (6.8%)	261 (26.5%)	310 (31.5%)	347 (35.2%)
Channel type								
Facebook	337 (34.2%)	99 (29.4%)	144 (42.7%)	94 (27.9%)	9 (2.7%)	44 (13.1%)	128 (38.0%)	156 (46.3%)
Twitter	197 (20.0%)	65 (33.0%)	82 (41.6%)	50 (25.4%)	6 (3.1%)	34 (17.3%)	67 (34.0%)	90 (45.7%)
YouTube	176 (17.9%)	56 (31.8%)	82 (46.6%)	38 (21.6%)	2 (1.1%)	33 (18.8%)	56 (31.8%)	85 (48.3%)
Instagram	134 (13.6%)	37 (27.6%)	62 (46.3%)	35 (26.1%)	6 (4.5%)	18 (13.4%)	33 (24.6%)	77 (57.5%)
Tik Tok	101 (10.3%)	29 (28.7%)	43 (42.6%)	29 (28.7%)	4 (4.0%)	15 (14.9%)	23 (22.8%)	59 (58.4%)
Television	703 (71.4%)	143 (20.3%)	323 (46.0%)	237 (33.7%)	46 (6.5%)	176 (25.0%)	238 (33.9%)	243 (34.6%)
Newspaper	261 (26.5%)	59 (22.6%)	113 (43.3%)	89 (34.1%)	17 (6.5%)	78 (29.9%)	92 (35.3%)	74 (28.4%)
Radio	119 (12.1%)	32 (26.9%)	40 (33.6%)	47 (39.5%)	10 (8.4%)	23 (19.3%)	45 (37.8%)	41 (34.5%)
Most information channel								
Not social media	767 (77.9%)	189(24.6%)	328 (42.8%)	250 (32.6%)	65 (8.5%)	241 (31.4%)	237 (30.9%)	224 (29.2%)
Social media	218 (22.1%)	65 (29.8%)	104 (47.7%)	49 (22.5%)	2 (0.9%)	20 (9.2%)	73 (33.5%)	123 (56.4%)

**Table 4 vaccines-10-02096-t004:** Multivariable Logistic Regression of Vaccine Hesitancy.

Variables	Odds Ratio of Vaccine Hesitancy Compared to Acceptance	(95% CI)
Using social media as the main source	1.50	(1.03, 2.19) *
Age		
18–24	1	Reference
25–34	0.91	(0.56, 1.48)
35–44	0.85	(0.53, 1.39)
45–54	0.51	(0.31, 0.85) *
Over 54	0.40	(0.21, 0.74) **
Provincial region		
Western †	1	Reference
Ontario	1.12	(0.71, 1.77)
Quebec	1.69	(1.09, 2.62) *
Atlantic ‡	1.64	(0.78, 3.44)
Northern ϕ	10.59	(1.16, 96.64) *
Other	0.79	(0.09, 7.06)
Race		
White	1	Reference
Black	1.30	(0.65, 2.59)
Asian	0.71	(0.42, 1.18)
Hispanic	1.90	(0.54, 6.72)
Other	1.47	(0.87, 2.48)
Education		
Less than high school	1	Reference
High school	0.86	(0.38, 1.94)
College/Bachelor’ degree	0.90	(0.41, 2.00)
Post-graduate	0.98	(0.41, 2.38)
Other	2.85	(0.25, 33.12)
Employment		
Working	1	Reference
Not working	1.68	(1.14, 2.49) **
Retired	0.55	(0.27, 1.11)
Other	0.79	(0.32, 1.97)
COVID-19 experience		
Diagnosed with COVID-19	1.50	(0.89, 2.52)
Concern for COVID-19		
Infecting others with COVID-19	0.45	(0.32, 0.63) ***
Trust in information		
Low	1	Reference
Some	0.25	(0.17, 0.36) ***
High	0.06	(0.04, 0.10) ***

† Western provinces: British Columbia, Alberta, Saskatchewan, and Manitoba. ‡ Atlantic provinces: Nova Scotia, New Brunswick, Prince Edward Island, and Newfoundland/Labrador. ϕ Northern provinces: Yukon, Northwest Territories, and Nunavut. * *p*-value <0.05; ** *p*-value <0.01; *** *p*-value <0.001.

## Data Availability

We have uploaded our data to the ICPSR COVID-19 Data Repository. The identifier is openicpsr-138562 and the URL is https://www.openicpsr.org/openicpsr/project/138562/version/V1/view (accessed on 30 July 2022).

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
