# Peer review of "A Descriptive Analysis of the Relationship between Social Media Use and Vaccine Hesitancy among a Sample of Unvaccinated Adults in Canada"

_vaccines, 2022, doi:10.3390/vaccines10122096_

Round 1
Reviewer 1 Report
Dear authors,
Many thanks for the opportunity to review your manuscript, which presents an interesting and timely study about the connections between social media and vaccine use.
I have a number of specific comments aimed at helping you to strengthen the paper:
1) In line 35 you mention June 2022, but end this sentence at line 38 with July 2022. The dissonance in dates here is confusing to the reader.
2) In line 88 you mention traditional and social media information channels. You define what you mean by each of these terms later in the paper (line 120-121 you discuss traditional media), but I think including a clear definition in the introductory section would be useful.
3) Line 111: Use of "raise", consider choosing a different word like cause
4) Line 143: Please specify what you mean by small monetary incentive- is this $.25 or $5 or $20?
5) In line 149 you mention English and French, which makes me wonder if the survey was offered in both languages? Please clarify.
6) You need to justify your sample of 985 respondents. This is critical to improving your paper. I'm not convinced 985 is significantly powered, and you do not address how you ended up with 985 responses. Is this merely a convenience sample?
7) I noticed that the Tables are cited out of order in the text and appear out of chronological order in the paper.
8) Lines 342-347: it would be helpful to expand this paragraph by suggesting how social media could be used to convince vaccine-hesitant populations. It feels like the paragraph is incomplete without a next step related to how to use the information you have presented.
9) Related to your sample, I am struggling with the provincial distribution in Table 3, and the lack of generalizability that you mention in lines 383-384. It is unclear to me if potentially the geographic distribution you ended up with in your sample somehow connects to the overall population distribution in Canada, or if it is a function of the survey? It begs the question as to why you did not aim for better geographic distribution in conducting the survey. Ultimately, I think this could be a major flaw in your design and results, given that your title suggests that the contents within are representative of Canada. They are not.
I hope these comments are useful as you revise your manuscript.
Author Response
Point 1: In line 35 you mention June 2022, but end this sentence at line 38 with July 2022. The dissonance in dates here is confusing to the reader.
Response 1: We are grateful for your advice. The dates have both been updated (lines 39, 42).
Point 2: In line 88 you mention traditional and social media information channels. You define what you mean by each of these terms later in the paper (line 120-121 you discuss traditional media), but I think including a clear definition in the introductory section would be useful.
Response 2: Thank you for your suggestion. An elaborated description has been added to the section (lines 93-94).
Point 3: Line 111: Use of "raise", consider choosing a different word like cause
Response 3: Thank you for your suggestion. Corrections have been made (line 124).
Point 4: Line 143: Please specify what you mean by small monetary incentive- is this $.25 or $5 or $20?
Response 4: Thank you for your advice. A more precise description has been added to the section (lines 158-161).
Point 5: In line 149 you mention English and French, which makes me wonder if the survey was offered in both languages? Please clarify.
Response 5: Thank you for your suggestion. An elaborated description has been added to the section (lines 183-191).
Point 6: You need to justify your sample of 985 respondents. This is critical to improving your paper. I'm not convinced 985 is significantly powered, and you do not address how you ended up with 985 responses. Is this merely a convenience sample?
Response 6: Thank you for this valuable advice. However, we note that this paper's intention is exploratory and descriptive, as indicated in the study's objectives, rather than seeking to make inferences about the prevalence and determinants of vaccine hesitancy in all of Canada, or even within a certain sub-group of Canadians. We have also modified the title to reflect the descriptive and exploratory nature of the study. Regarding the concern about the minimal sample size, we did not plan to test any a priori hypotheses that may necessitate sample size or power calculations. The findings in this study are exploratory and descriptive in that they outline the factors that influence vaccine hesitancy in our sample. Also, there were also some studies exploring the association between social media engagement and vaccine hesitancy during the COVID-19 pandemic, with a sample size of 1050 and 1012 [[1],[2]]. We inclined to believe that our sample size of 985 was appropriate.
Point 7: I noticed that the Tables are cited out of order in the text and appear out of chronological order in the paper.
Response 7: Thank you for your advice. Corrections have been made.
Point 8: Lines 342-347: it would be helpful to expand this paragraph by suggesting how social media could be used to convince vaccine-hesitant populations. It feels like the paragraph is incomplete without a next step related to how to use the information you have presented.
Response 8: Thank you for this valuable advice, which will greatly improve our work. We expanded the paragraph by suggesting how social media could be used to promote vaccine uptake in vaccine-hesitant populations (lines 419-438).
Point 9: Related to your sample, I am struggling with the provincial distribution in Table 3, and the lack of generalizability that you mention in lines 383-384. It is unclear to me if potentially the geographic distribution you ended up with in your sample somehow connects to the overall population distribution in Canada, or if it is a function of the survey? It begs the question as to why you did not aim for better geographic distribution in conducting the survey. Ultimately, I think this could be a major flaw in your design and results, given that your title suggests that the contents within are representative of Canada. They are not.
Response 9: Thank you for your constructive comments. As stated above, the intention of this study is exploratory and descriptive rather than seeking to make inferences about the prevalence and determinants of vaccine hesitancy in all of Canada, or even within a certain sub-group of Canadians. We modified the title to reflect the descriptive and exploratory nature of the study: “A descriptive analysis of the relationship between social media use and vaccine hesitancy among a sample of unvaccinated adults in Canada.”
[1] Al-Uqdah, L., Franklin, F. A., Chiu, C. C., & Boyd, B. N. (2022). Associations Between Social Media Engagement and Vaccine Hesitancy. Journal of community health, 47(4), 577–587. https://doi.org/10.1007/s10900-022-01081-9
[2] Viswanath, K., Bekalu, M., Dhawan, D., Pinnamaneni, R., Lang, J., & McLoud, R. (2021). Individual and social determinants of COVID-19 vaccine uptake. BMC public health, 21(1), 818. https://doi.org/10.1186/s12889-021-10862-1
Reviewer 2 Report
Thank you for the invitation to review this manuscript. I found this study interesting but have some concerns which should be considered before making any decision regarding the fate of this manuscript.
1. The authors have used roman in-text citations. This is the first time I have seen such citation style. Please follow the Numerics.
2. In the presence of various investigations from Canada, how this study differs from others? There is a need to provide a stance on how the current study covers the literature gap.
3. The authors have stated that small incentives were provided to the participants. Can authors explain a bit more about such incentives?
4. For VH, the authors have only considered very likely response to indicate whether the respondent is vaccine acceptor or not? What about the response "somewhat likely"? It means that the participants opting option somewhat likely were considered as vaccine hesitant. I am not sure how the authors came up with such an approach. I think the respondents somewhat likely (14% in this study) are not vaccine hesitant.
5. The authors did not provide any information on the mechanism of validation, reliability analysis and translation of the data collection form.
6. Please clarify that the data collection form was available both in English and French languages. The authors are encouraged to provide data collection as supplementary files in both languages.
7. It would be more helpful for the readers if the authors could provide the construct of data collection form i.e. how many questions and sections were present in the study instrument.
8. The structure of the tables needs improvement. The tables are rich with large statements. It's not clear why some rows and tables are highlighted.
9. Why was VH higher in northern provinces?
10. The authors did not provide reasoning for factors associated with the VH or vaccine acceptance in the discussion section. The authors are suggested to compare all the results with studies already conducted in Canada as well as other parts of the world. The results' discussion hardly covers one page of the manuscript. Various demographic features were found to be associated with VH, but they were not discussed and compared with other studies in the discussion section.
Author Response
Point 1: The authors have used roman in-text citations. This is the first time I have seen such citation style. Please follow the Numerics.
Response 1: Thank you for the advice. Corrections have been made.
Point 2: In the presence of various investigations from Canada, how this study differs from others? There is a need to provide a stance on how the current study covers the literature gap.
Response 2: Thank you for this valuable advice. We have added the description that there is limited data assessing the impact of traditional and social media information channels on the public's attitude toward the COVID-19 vaccine in Canada. (lines 139-140)
Point 3: The authors have stated that small incentives were provided to the participants. Can authors explain a bit more about such incentives?
Response 3: Thank you for your advice. A more precise description has been added to the section (lines 158-161).
Point 4: For VH, the authors have only considered very likely response to indicate whether the respondent is vaccine acceptor or not? What about the response "somewhat likely"? It means that the participants opting option somewhat likely were considered as vaccine hesitant. I am not sure how the authors came up with such an approach. I think the respondents somewhat likely (14% in this study) are not vaccine hesitant.
Response 4: Thank you for your constructive comments. As vaccine hesitancy is defined by World Health Organization as “delay in acceptance or refusal of vaccines despite availability of vaccine services,” we considered the response "somewhat likely" as some level of hesitancy when we dichotomized the responses into vaccine-acceptant and vaccine-hesitant attitudes. Therefore, only respondents who reported being willing to take the vaccine without any hesitation fell under vaccine-acceptant category. Furthermore, we are unable to validate our findings in terms of the impact of the variables we discovered to be associated with intention to get vaccinated compared to actual behaviors because we are unsure of the degree to which expressed intent to take a vaccine is associated with actual behavior. We have further included this as a limitation. (lines 450-453)
Point 5: The authors did not provide any information on the mechanism of validation, reliability analysis and translation of the data collection form.
Response 5: Thank you for your comments. We have included a more precise description of the mechanism of validation and translation of the data collection form. (lines 184-194)
Point 6: Please clarify that the data collection form was available both in English and French languages. The authors are encouraged to provide data collection as supplementary files in both languages.
Response 6: Thank you for this valuable advice. A more precise description has been added to the section Data Collection. The English version and French version of the questionnaire are provided as Supplementary Material File S1 to this manuscript. (lines 183-191)
Point 7: It would be more helpful for the readers if the authors could provide the construct of data collection form i.e. how many questions and sections were present in the study instrument.
Response 7: Thank you for the advice. A more precise description has been added to the section Data Collection, along with Supplementary Material File provided. (lines 183-194)
Point 8: The structure of the tables needs improvement. The tables are rich with large statements. It's not clear why some rows and tables are highlighted.
Response 8: Thank you for your suggestions. Corrections have been made.
Point 9: Why was VH higher in northern provinces?
Response 9: Thank you for your comments. In fact, the interprovincial comparisons are challenging due to the number of participants was small in some categories (eg, Atlantic and Northern provinces). For example, there were only 11 participants from the northern provinces (10 hesitant and 1 acceptant). Although living in the Northern provinces had an odds ratio point estimate of 10.59, this finding was based on only one vaccine-acceptant respondent and had wide confidence intervals. Therefore, the interprovincial difference is hardly interpretable, and this is recognized as a limitation of our study (lines 445-447).
Point 10: The authors did not provide reasoning for factors associated with the VH or vaccine acceptance in the discussion section. The authors are suggested to compare all the results with studies already conducted in Canada as well as other parts of the world. The results' discussion hardly covers one page of the manuscript. Various demographic features were found to be associated with VH, but they were not discussed and compared with other studies in the discussion section.
Response 10: Thank you for this valuable advice, which will greatly improve our work. We have expanded the relevant paragraphs in the Discussion section, going into greater depth on how the results of this study compare to those of previous studies: With regard to sociodemographic factors, we found respondents aged 45-54 years and over 54 years had significantly decreased odds of being vaccine hesitant when compared to those aged 18-24 years. Our results are consistent with previous studies in Canada and also align with trends in the United States, which indicate that younger individuals (those under 65 years old) are less acceptant to the COVID-19 vaccine [[1]]. Dzieciolowska et al. found age over 50 to be determinants of increased vaccine acceptance among healthcare workers in Canada [[2]]. Ogilvie et al. reported that individuals above the age of 60 were more likely to be vaccine acceptant [[3]]. On the contrary, Tang et al. indicated that being aged 40-59 was associated with lower vaccination intention [[4]]. In general, younger populations are less willing to receive the COVID-19 vaccine, which may in part be explained by the way they obtain information and make health decisions. To promote vaccine acceptance in this population, it is imperative to have more targeted information which reassures them that the vaccine is both necessary and safe. (lines 385-397)
[1] Szilagyi, P. G., Thomas, K., Shah, M. D., Vizueta, N., Cui, Y., Vangala, S., & Kapteyn, A. (2020). National Trends in the US Public's Likelihood of Getting a COVID-19 Vaccine-April 1 to December 8, 2020. JAMA, 325(4), 396–398. Advance online publication. https://doi.org/10.1001/jama.2020.26419
[2] Dzieciolowska, S., Hamel, D., Gadio, S., Dionne, M., Gagnon, D., Robitaille, L., Cook, E., Caron, I., Talib, A., Parkes, L., Dubé, È., & Longtin, Y. (2021). Covid-19 vaccine acceptance, hesitancy, and refusal among Canadian healthcare workers: A multicenter survey. American journal of infection control, 49(9), 1152–1157. https://doi.org/10.1016/j.ajic.2021.04.079
[3] Ogilvie, G. S., Gordon, S., Smith, L. W., Albert, A., Racey, C. S., Booth, A., Gottschlich, A., Goldfarb, D., Murray, M., Galea, L., Kaida, A., Brotto, L. A., & Sadarangani, M. (2021). Intention to receive a COVID-19 vaccine: results from a population-based survey in Canada. BMC public health, 21(1), 1017. https://doi.org/10.1186/s12889-021-11098-9
[4] Tang, X., Gelband, H., Nagelkerke, N., Bogoch, I. I., Brown, P., Morawski, E., Lam, T., Jha, P., & Action to beat coronavirus/Action pour battre le coronavirus (Ab-C) Study Investigators (2021). COVID-19 vaccination intention during early vaccine rollout in Canada: a nationwide online survey. Lancet Regional Health. Americas, 2, 100055. https://doi.org/10.1016/j.lana.2021.100055
Reviewer 3 Report
This is a very informative paper and it provides a comprehensive insight into an important topic. However, some issues need to be addressed:- Lines 23-24: Do you mean versus those who did not use social media as their primary source? Clarify, rephrase slightly.
- Lines 43-44: It would be illustrative to give figures here for the rates in Canada and the US and Europe.
- The Introduction section is very well written and provides a comprehensive overview of the topic. However, I would recommend to the authors to consider revising the text in this section so that some of the parts of Introduction are incorporated in the section Discussion (e.g. text at lines 67-75, etc.) - as it will be useful to discuss individual studies at this level once comparing your own results to that previously published, while in this section you should sum up the findings. For example, do not dedicate one or more sentences to each study finding that male sex was a predictor of higher intentions to get vaccinated, but rather list the predictors identified in the studies and then cite multiple studies after each indicator that have found it. Then also cite a few that did not or that had opposite results - this explains the knowledge gap that you will be addressing.
- Lines 87-89: This sentence could be moved closer to the end of the section Introduction, where you explain why you conducted the study and its aims.
- Line 151: Given the time when the study was conducted, add a description of the vaccination process status in Canada at that point. When did vaccines become available? What were the procedures for vaccination, who was eligible to get the vaccinations at that time? What were the recommendations and guidelines, was it mandatory etc. Describe the general vaccination situation setting in which you are asking participants about their intention to get vaccinated against COVID-19.
- Lines 151-153: Did inclusion criteria cover whether participants had COVID-19?
- Line 153: How do you define fully vaccinated at that point in Canada?
- Lines 164-165: You decided that the answer "somewhat likely" does not correspond with vaccine-acceptant attitude? Could this have influenced the results if that answer was considered in that category?
- Line 174: A part of the section Introduction was dedicated to explaining how vaccine-critical websites and blogs reduced vaccine acceptance. You did not investigate these as sources of information, only social media?
- Lines 188-189: Why were workers considered as working in healthcare vs not working in the healthcare sector only? What was the situation then, were any other professions considered as primary for receiving vaccines, for example teachers (as you have also mentioned them in the section Introduction).
- Lines 188-189: What was the rationale behind not observing three categories for level of risk here, but two instead?
- Methods: Provide rationale for the sample size calculations. It is important to see whether the study had enough power to identify meaningful differences based on the results of previous studies (e.g. previously reported prevalence of hesitancy). Provide citation for that too.
- Line 230: What does reference No. 40 mean here?
- Table 1: Consider adding another row to the first row, above the not social media and social media - to indicate that this means "primary source of COVID-19 vaccination information".
- Line 250: Rename this section, as there were descriptive statistics presented earlier too. Be specific in naming the section, what does data refer to?
- Results: Why is Table 2 third and Table 3 second? Both in citing and describing in the text, as well as in order of appearance. Reorder.
- Results: In general, the section is very informative, however there is a lot of text that simply repeats the information presented in the Tables (that are very well composed and comprehensive), so consider describing the most important and significant results in the text only, rather than repeating information from the tables.
- Lines 322-324: This sentence does not belong to the first paragraph of the section Discussion, where you sum up your findings, rather move it towards the end of the section where you conclude and describe implications of your results.
- Lines 369-371: Don't these platforms have in place algorithms and mechanisms that warn the users regarding COVID-19 information that is placed, as well as remove content that is considered not reputable?
- Discussion: Introduction is more detailed than this section, and that should not be the case. Revise accordingly, and discuss in more detail the findings of this study as they compare to results of other studies, with providing possible explanations, in particular with regard to sociodemographic factors.
Author Response
Point 1: Lines 23-24: Do you mean versus those who did not use social media as their primary source? Clarify, rephrase slightly.
Response 1: Thank you for this valuable advice. Corrections have been made (lines 27-28).
Point 2: Lines 43-44: It would be illustrative to give figures here for the rates in Canada and the US and Europe.
Response 2: Thank you for your constructive suggestions. We have included Figure 1, which demonstrates the cumulative confirmed COVID-19 cases and deaths in Canada, the United States and Europe.
Figure 1. Cumulative confirmed COVID-19 cases and deaths in Canada, the United States and Europe prior to November 2021. (Accessed through Our world in data [OurWorldInData.org]. Available from: https://ourworldindata.org/covid-vaccinations. Source: Johns Hopkins University CSSE COVID-19 Data.)
Point 3: The Introduction section is very well written and provides a comprehensive overview of the topic. However, I would recommend to the authors to consider revising the text in this section so that some of the parts of Introduction are incorporated in the section Discussion (e.g. text at lines 67-75, etc.) - as it will be useful to discuss individual studies at this level once comparing your own results to that previously published, while in this section you should sum up the findings. For example, do not dedicate one or more sentences to each study finding that male sex was a predictor of higher intentions to get vaccinated, but rather list the predictors identified in the studies and then cite multiple studies after each indicator that have found it. Then also cite a few that did not or that had opposite results - this explains the knowledge gap that you will be addressing.
Response 3: Thank you for this valuable advice, which will greatly improve our work. We have edited the sections Introduction and Discussion accordingly (lines 60-66, 139-140, 385-397).
Point 4: Lines 87-89: This sentence could be moved closer to the end of the section Introduction, where you explain why you conducted the study and its aims.
Response 4: Thank you for the advice. Corrections have been made (line 139-140).
Point 5: Line 151: Given the time when the study was conducted, add a description of the vaccination process status in Canada at that point. When did vaccines become available? What were the procedures for vaccination, who was eligible to get the vaccinations at that time? What were the recommendations and guidelines, was it mandatory etc. Describe the general vaccination situation setting in which you are asking participants about their intention to get vaccinated against COVID-19.
Response 5: Thank you for your suggestions. A description has been added: “After the first COVID-19 vaccine was authorized for use in Canada on December 9, 2020, healthcare workers and residents of long-term care institutions were given priority to be immunized [[1]]. Based on national guidance available at the time the survey was developed, populations with mortality risk factors, such as advanced age and comorbidities, were also included in the vaccine priority group [[2]]. Some provinces further gave priority to people who were at a higher risk of contracting the disease, such as essential frontline workers and residents of areas that were severely affected [[3]].” (lines 168-175)
Point 6: Lines 151-153: Did inclusion criteria cover whether participants had COVID-19?
Response 6: Thank you for your comments. Our inclusion criteria did not cover whether participants had COVID-19. Users were eligible to participate the survey if they were over 18 years of age and had not yet been fully vaccinated at the time the survey was administered.
Point 7: Line 153: How do you define fully vaccinated at that point in Canada?
Response 7: Thank you for your comments. The fully vaccinated individuals were defined as those who completed the initial protocol of COVID-19 vaccines (2 doses of an accepted COVID-19 vaccine, a mix of 2 accepted vaccines, or one dose of the Janssen/Johnson & Johnson vaccine). The elaborated description has been added to the section (lines 177-178).
Point 8: Lines 164-165: You decided that the answer "somewhat likely" does not correspond with vaccine-acceptant attitude? Could this have influenced the results if that answer was considered in that category?
Response 8: Thank you for your constructive comments. As vaccine hesitancy is defined by World Health Organization as “delay in acceptance or refusal of vaccines despite availability of vaccine services,” we considered the response "somewhat likely" as some level of hesitancy when we dichotomized the responses into vaccine-acceptant and vaccine-hesitant attitudes. Therefore, only respondents who reported being willing to take the vaccine without any hesitation fell under vaccine-acceptant category. Furthermore, we are unable to validate our findings in terms of the impact of the variables we discovered to be associated with intention to get vaccinated compared to actual behaviors because we are unsure of the degree to which expressed intent to take a vaccine is associated with actual behavior. We have further included this as a limitation (line 450-453). Indeed, if we included "somewhat likely " to the vaccine-acceptant category, it is possible that the statistical efficiency would be lower since the sample sizes in each category would be less evenly distributed, and this could lead to results that were not statistically significant.
Point 9: Line 174: A part of the section Introduction was dedicated to explaining how vaccine-critical websites and blogs reduced vaccine acceptance. You did not investigate these as sources of information, only social media?
Response 9: Thank you for your comments. Although anti-vaccine websites and blogs can influence vaccine uptake, we focused our research on the impact of social media due to its interactive nature. Social media allows users to quickly create and share material globally without requiring editorial oversight, scientific review, or evidence-based evaluation. Direct access to the public, networking with media and other organizations, the capacity to communicate with and comprehend users, and the ability to further evaluate campaigns make social media a unique information platform. In our survey, we asked the participants to select up to three sources from which they received the most information regarding the COVID-19 vaccine. The options included government agency websites and news portal websites like Yahoo! or MSN. However, those sources could also be promoted or shared on social media channels. We sought to highlight the effect of using social media versus never using it as the primary source of COVID-19 vaccine information. The aforementioned influence of the interchange of different sources and the cumulative effect of multiple sources could not be assessed in our analysis and was recognized as a limitation.
Point 10: Lines 188-189: Why were workers considered as working in healthcare vs not working in the healthcare sector only? What was the situation then, were any other professions considered as primary for receiving vaccines, for example teachers (as you have also mentioned them in the section Introduction).
Response 10: Thank you for your comments. Healthcare workers and residents of long-term care facilities were prioritized to be immunized according to national guidance available at the time the survey was developed.[4] It is true that some other professions were also considered as primary for receiving vaccines, such as teachers, given the importance of minimizing disruptions to education and the fact that COVID-19 vaccines had not yet been approved for children at that time. However, the initial phase of COVID-19 vaccination campaign focused primarily targeted residents and staff of long-term care facilities and health-care workers. In addition, since health-care workers are also viewed as role models by the general public, their endorsement of vaccination could eventually change the attitudes towards vaccines among hesitant individuals.[5]
Point 11: Lines 188-189: What was the rationale behind not observing three categories for level of risk here, but two instead?
Response 11: Thank you for your comments. The way we have constructed risk perception variables is having each item as an indicator. Therefore, we dichotomized the responses into two risk-related categories (medium/low reported risk versus high reported risk).
Point 12: Methods: Provide rationale for the sample size calculations. It is important to see whether the study had enough power to identify meaningful differences based on the results of previous studies (e.g. previously reported prevalence of hesitancy). Provide citation for that too.
Response 12: Thank you for this valuable advice. However, we note that this paper's intention is exploratory and descriptive, as indicated in the study's objectives, rather than seeking to make inferences about the prevalence and determinants of vaccine hesitancy in all of Canada, or even within a certain sub-group of Canadians. We have also modified the title to reflect the descriptive and exploratory nature of the study. Regarding the concern about the minimal sample size, we did not plan to test any a priori hypotheses that may necessitate sample size or power calculations. The findings in this study are exploratory and descriptive in that they outline the factors that influence vaccine hesitancy in our sample. Also, there were also some studies exploring the association between social media engagement and vaccine hesitancy during the COVID-19 pandemic, with a sample size of 1050 and 1012 [[6],[7]]. We inclined to believe that our sample size of 985 was appropriate.
Point 13: Line 230: What does reference No. 40 mean here?
Response 13: Thank you for your comments. It was intended to provide a reference for the data analysis software we utilized. We removed the reference to prevent confusion (line 257).
Point 14: Table 1: Consider adding another row to the first row, above the not social media and social media - to indicate that this means "primary source of COVID-19 vaccination information".
Response 14: Thank you for the advice. Corrections have been made (Table 1).
Point 15: Line 250: Rename this section, as there were descriptive statistics presented earlier too. Be specific in naming the section, what does data refer to?
Response 15: Thank you for the advice. We have renamed the section “Descriptive statistics of COVID-19 Information from Different Channels.” (line 286)
Point 16: Results: Why is Table 2 third and Table 3 second? Both in citing and describing in the text, as well as in order of appearance. Reorder.
Response 16: Thank you for the advice. Corrections have been made.
Point 17: Results: In general, the section is very informative, however there is a lot of text that simply repeats the information presented in the Tables (that are very well composed and comprehensive), so consider describing the most important and significant results in the text only, rather than repeating information from the tables.
Response 17: Thank you for your valuable suggestions. we have trimmed some descriptions in the section accordingly (lines 262-266, 337-341).
Point 18: Lines 322-324: This sentence does not belong to the first paragraph of the section Discussion, where you sum up your findings, rather move it towards the end of the section where you conclude and describe implications of your results.
Response 18: Thank you for the advice. Corrections have been made (lines 424-426).
Point 19: Lines 369-371: Don't these platforms have in place algorithms and mechanisms that warn the users regarding COVID-19 information that is placed, as well as remove content that is considered not reputable?
Response 19: Thank you for your comments. These social media platforms generally remove or restrict the distribution of content that contests the accuracy of medical information or spreads false health claims that are deemed detrimental, such as inaccurate information regarding the risks associated with vaccinations.[8] Twitter, for example, manages its fact checking internally. On the contrary, Facebook and YouTube rely on collaborations with external fact checkers. These judgments are controversial in part because of how social media platforms define the nebulous terms "misinformation" or "disinformation," which rely on the notion of a scientific consensus. According to Sander van der Linden, professor of social psychology in society at Cambridge University, UK, rather than simply stating something as true or false, the platform should really connect users to content that educates them about the scientific method in order to build an appreciation of how science works. Therefore, we suggest that public health agencies work closely with social media companies in order to play a significant role in fact-checking COVID-19 vaccine misinformation. For example, they should work to create posts that will more effectively engage the public, since they are regarded as authoritative, reputable information sources and are likely to cause attitude and behavior changes regarding vaccine endorsement.[9]
Point 20: Discussion: Introduction is more detailed than this section, and that should not be the case. Revise accordingly, and discuss in more detail the findings of this study as they compare to results of other studies, with providing possible explanations, in particular with regard to sociodemographic factors.
Response 20: Thank you for this valuable advice, which will greatly improve our work. We have expanded the section: “With regard to sociodemographic factors, we found respondents aged 45-54 years and over 54 years had significantly decreased odds of being vaccine hesitant when compared to those aged 18-24 years. Our results are consistent with previous studies in Canada and also align with trends in the United States, which indicate that younger individuals (those under 65 years old) are less acceptant to the COVID-19 vaccine [[10]]. Dzieciolowska et al. found age over 50 to be determinants of increased vaccine acceptance among healthcare workers in Canada [[11]]. Ogilvie et al. reported that individuals above the age of 60 were more likely to be vaccine acceptant [[12]]. On the contrary, Tang et al. indicated that being aged 40-59 was associated with lower vaccination intention [[13]]. In general, younger populations are less willing to receive the COVID-19 vaccine, which may in part be explained by the way they obtain information and make health decisions. To promote vaccine acceptance in this population, it is imperative to have more targeted information which reassures them that the vaccine is both necessary and safe.” (lines 385-397)
[1] National Advisory Committee on Immunization (NACI). Guidance on the Prioritization of Key Populations for COVID-19 Immunization. Ottawa: Government of Canada; 2020.
[2] Government of Canada. Vaccines for COVID-19: Shipments and deliveries Canada.ca [Internet]. 2021 ND [cited 2021 Apr 27]. Available from: https://www.canada.ca/en/public- health/services/diseases/2019- novelcoronavirus-infection/prevention-risks/covid-19-vaccine-treatment/vaccine-rollout.html
[3] Ministry of Health. COVID-19 vaccines for Ontario COVID-19 (coronavirus) in Ontario [Internet]. 2021 ND [cited 2021 Apr 27]. Available from: https://covid-19.ontario.ca/covid-19-vaccines-ontario
[4] National Advisory Committee on Immunization (NACI). Guidance on the Prioritization of Key Populations for COVID-19 Immunization. Ottawa: Government of Canada; 2020.
[5] Yaqub O, Castle-Clarke S, Sevdalis N, Chataway J. Attitudes to vaccination: a critical review. Soc Sci Med. 2014;112:1–11.
[6] Al-Uqdah, L., Franklin, F. A., Chiu, C. C., & Boyd, B. N. (2022). Associations Between Social Media Engagement and Vaccine Hesitancy. Journal of community health, 47(4), 577–587. https://doi.org/10.1007/s10900-022-01081-9
[7] Viswanath, K., Bekalu, M., Dhawan, D., Pinnamaneni, R., Lang, J., & McLoud, R. (2021). Individual and social determinants of COVID-19 vaccine uptake. BMC public health, 21(1), 818. https://doi.org/10.1186/s12889-021-10862-1
[8] Clarke L. (2021). Covid-19: Who fact checks health and science on Facebook?. BMJ (Clinical research ed.), 373, n1170. https://doi.org/10.1136/bmj.n1170
[9] Xue, H., Gong, X., & Stevens, H. (2022). COVID-19 Vaccine Fact-Checking Posts on Facebook: Observational Study. Journal of medical Internet research, 24(6), e38423. https://doi.org/10.2196/38423
[10] Szilagyi, P. G., Thomas, K., Shah, M. D., Vizueta, N., Cui, Y., Vangala, S., & Kapteyn, A. (2020). National Trends in the US Public's Likelihood of Getting a COVID-19 Vaccine-April 1 to December 8, 2020. JAMA, 325(4), 396–398. Advance online publication. https://doi.org/10.1001/jama.2020.26419
[11] Dzieciolowska, S., Hamel, D., Gadio, S., Dionne, M., Gagnon, D., Robitaille, L., Cook, E., Caron, I., Talib, A., Parkes, L., Dubé, È., & Longtin, Y. (2021). Covid-19 vaccine acceptance, hesitancy, and refusal among Canadian healthcare workers: A multicenter survey. American journal of infection control, 49(9), 1152–1157. https://doi.org/10.1016/j.ajic.2021.04.079
[12] Ogilvie, G. S., Gordon, S., Smith, L. W., Albert, A., Racey, C. S., Booth, A., Gottschlich, A., Goldfarb, D., Murray, M., Galea, L., Kaida, A., Brotto, L. A., & Sadarangani, M. (2021). Intention to receive a COVID-19 vaccine: results from a population-based survey in Canada. BMC public health, 21(1), 1017. https://doi.org/10.1186/s12889-021-11098-9
[13] Tang, X., Gelband, H., Nagelkerke, N., Bogoch, I. I., Brown, P., Morawski, E., Lam, T., Jha, P., & Action to beat coronavirus/Action pour battre le coronavirus (Ab-C) Study Investigators (2021). COVID-19 vaccination intention during early vaccine rollout in Canada: a nationwide online survey. Lancet Regional Health. Americas, 2, 100055. https://doi.org/10.1016/j.lana.2021.100055

Round 2
Reviewer 1 Report
Dear authors, thank you for this thoughtful revision of your manuscript. I think that the update to the article title, in particular, is very helpful in clarifying the overall intent of the study. I wish you luck with this manuscript!
Author Response
Response: We feel great thanks for your professional review work on our article. All of your comments and suggestions have contributed a lot to improve the quality of our study. Thank you again for the time and effort you have put into your comments.

Reviewer 3 Report
I would like to thank the Authors for the revisions they have made and for addressing most of my comments. The revisions they implemented have significantly improved the manuscript. It appears more coherent, sections are easier to read and more logical, and clarifications were made where necessary, due to the additional explanations that were provided. The following remarks remain:
- Line 48: The Figure mentioned here cannot be found in the supplied manuscript file. However, regarding your response No. 2 - I have in fact meant to add numbers (figures), e.g. rates etc. in the text. - Regarding your response No. 6 - since you did not ask the participants whether they had COVID-19, it would seem important to note if there were, at the time when the study was conducted, any recommendations or rules for vaccination of persons who have had COVID-19? If so, this might be a limitation. - Lines 396-398: Consider expanding this sentence slightly, by using the response from the Response letter where you explained that social networks should not simply mark content as misinformation or fake but also provide explanations and educate the user with trustworthy information. Your response No. 19 provides a great source for this.Author Response
Point 1: Line 48: The Figure mentioned here cannot be found in the supplied manuscript file. However, regarding your response No. 2 - I have in fact meant to add numbers (figures), e.g. rates etc. in the text.
Response 1: Thank you for your valuable suggestions. We have included Figure 1 in the text and made modifications to our manuscript accordingly: “The prevalence and mortality rates of COVID-19 in Canada were generally lower when compared to the United States and European countries prior to November 2021 (cumulative cases per million people by November 1st 2021, US, 136,454, Europe, 87,865, Canada, 44,811; cumulative deaths per million people by November 1st 2021, US, 2,206, Europe, 1,798, Canada, 755; Figure 1) [[1],[2]]. However, a substantial increase in cases after that time highlighted the need for continued public health response and prevention efforts.” (lines 46-52)
Point 2: Regarding your response No. 6 - since you did not ask the participants whether they had COVID-19, it would seem important to note if there were, at the time when the study was conducted, any recommendations or rules for vaccination of persons who have had COVID-19? If so, this might be a limitation.
Response 2: Thank you again for your constructive comments. We asked the participants whether they had COVID-19, and we included "having been diagnosed with COVID-19" as a covariate in our analysis. Echoing the findings of previous studies that having a prior COVID-19 diagnosis is associated with higher COVID-19 vaccine hesitancy [[3],[4]], our results showed that a higher proportion of those with prior COVID-19 diagnosis (64.7%) were hesitant in taking the vaccine. This might be explained by a paucity of knowledge on the length of immunity provided by infection, as well as recommendations or guidelines for vaccination of those who have had COVID-19. We adjusted for this potential predictor of vaccine hesitancy in our regression model (Table 2 and Table 4). However, our analysis relied on self-reports of COVID-19, which may not be reliable, rather than cases that had been verified in a laboratory. The condition mentioned above has also been included as a limitation. (lines 417-420)
Point 3: Lines 396-398: Consider expanding this sentence slightly, by using the response from the Response letter where you explained that social networks should not simply mark content as misinformation or fake but also provide explanations and educate the user with trustworthy information. Your response No. 19 provides a great source for this.
Response 3: Thank you for your positive comments and valuable suggestions to improve the quality of our manuscript. We have made modifications accordingly (lines 405-408): “Rather than simply marking content as false or misinformation and then removing or restricting the distribution of content, there is also an opportunity for these social networks to provide explanations and enlighten the users with reliable information [[5],[6]].”
[1] National Collaborating Centre for Infectious Diseases. [cited 2022 July 30]. https://nccid.ca/covid-19-variants/
[2] Accessed through Our world in data [OurWorldInData.org]. [cited 2022 November 30]. Available from: https://ourworldindata.org/covid-vaccinations. Source: Johns Hopkins University CSSE COVID-19 Data.
[3] Reno, C., Maietti, E., Fantini, M. P., Savoia, E., Manzoli, L., Montalti, M., & Gori, D. (2021). Enhancing COVID-19 Vaccines Acceptance: Results from a Survey on Vaccine Hesitancy in Northern Italy. Vaccines, 9(4), 378. https://doi.org/10.3390/vaccines9040378
[4] Nguyen, K. H., Huang, J., Mansfield, K., Corlin, L., & Allen, J. D. (2022). COVID-19 Vaccination Coverage, Behaviors, and Intentions among Adults with Previous Diagnosis, United States. Emerging infectious diseases, 28(3), 631–638. https://doi.org/10.3201/eid2803.211561
[5] Clarke L. (2021). Covid-19: Who fact checks health and science on Facebook?. BMJ (Clinical research ed.), 373, n1170. https://doi.org/10.1136/bmj.n1170
[6] Xue, H., Gong, X., & Stevens, H. (2022). COVID-19 Vaccine Fact-Checking Posts on Facebook: Observational Study. Journal of medical Internet research, 24(6), e38423. https://doi.org/10.2196/38423
